# Peptide-Mediated Targeted Delivery of Aloe-Emodin as Anticancer Drug

**DOI:** 10.3390/molecules27144615

**Published:** 2022-07-19

**Authors:** Annarita Stringaro, Stefano Serra, Alessandro Gori, Annarica Calcabrini, Marisa Colone, Maria Luisa Dupuis, Francesca Spadaro, Serena Cecchetti, Alberto Vitali

**Affiliations:** 1National Center for Drug Research and Evaluation, Istituto Superiore di Sanità, Viale Regina Elena, 299, 00161 Rome, Italy; marisa.colone@iss.it (M.C.); marialuisa.dupuis@iss.it (M.L.D.); 2CNR-Institute of Chemical Sciences and Technologies “G. Natta” (SCITEC), 20131 Milan, Italy; stefano.serra@cnr.it (S.S.); alessandro.gori@cnr.it (A.G.); 3Microscopy Unit and Core Facilities, Istituto Superiore di Sanità, Viale Regina Elena, 299, 00161 Rome, Italy; francesca.spadaro@iss.it (F.S.); serena.cecchetti@iss.it (S.C.); 4CNR-Institute of Chemical Sciences and Technologies “G. Natta” (SCITEC), 00168 Rome, Italy

**Keywords:** bioconjugate, Aloe-emodin, breast cancer, SKBR3, HER2, drug delivery

## Abstract

Breast cancer is one of the most diffuse cancers in the world and despite the availability of the different drugs employed against it, the need for new and particularly more specific molecules is ever growing. In this framework, natural products are increasingly assuming an important role as new anticancer drugs. Aloe-emodin (AE) is one of the best characterized molecules in this field. The functionalization of bioactive natural products with selected peptide sequences to enhance their bioavailability and specificity of action is a powerful and promising strategy. In this study, we analyzed the cell specificity, cell viability effects, intracellular distribution, and immune cell response of a new peptide conjugate of Aloe-emodin in SKBR3 and A549 cell lines by means of viability tests, flow cytometry, and confocal microscopy. The conjugate proved to be more effective at reducing cell viability than AE in both cell lines. Furthermore, the results showed that it was mainly internalized within the SKBR3 cells, showing a nuclear localization, while A459 cells displayed mainly a cytoplasmic distribution. A preserving effect of the conjugate on NKs’ cell function was also observed. The designed conjugate showed a promising specific activity towards HER2-expressing cells coupled with an enhanced water solubility and a higher cytotoxicity; thus, the resulting proof-of-concept molecule can be further improved as an anticancer compound.

## 1. Introduction

Breast cancer is currently the most prevalent cancer in the world, as approximately 8 million women have been diagnosed in the last 5 years. In 2020, there were 2.3 million women diagnosed with breast cancer and about 700,000 deaths globally. (https://www.iarc.who.int/search/breast%20cancer, accessed on 1 May 2022). Breast cancer remains the second leading cause of cancer deaths among women [1]. Antibody drugs such as Cetuximab and Panitumumab are currently among the most widely used anticancer drugs against breast cancer, but their cost is a major problem.

The design of pharmacologically active molecules that are highly selective against cancer cells, but at same time possess affordable production costs, is therefore one of the major commitments of current research for the development of innovative and possibly affordable drugs. The delivery of drugs within specific cells is often a difficult task to achieve and has proven to be a major challenge for academic and industrial researchers.

One of the main obstacles encountered by using natural compounds as drugs is represented by their poor water solubility, which drastically reduces their bioavailability and hence their efficacy. In particular, it emerges from the literature that many molecules of natural origins demonstrate an enormous therapeutic potential especially if combined with efficient delivery systems [2,3]. The specific or targeted delivery of drugs into cancer cells, to increase their efficacy and decrease their cytotoxicity towards normal cells, involves (i) the use of a biological carrier that can be covalently bound to a therapeutic agent forming a so-called “conjugate”, or (ii) the encapsulation of the drugs in a special polymer that can be suitably functionalized to make it specific towards certain cell populations. The chemical conjugation of biomolecules with antibodies or selected peptide sequences is a well acknowledged procedure [4,5].

In this sense, the human epidermal growth factor receptor 2 (HER2, HER2/neu, and ErbB2) represents one of the most important targets for drug addressing or delivery in cancer therapy [6]. The overexpression of HER2 occurs mainly in invasive breast cancers (approximately of 15–30%) and gastric/gastroesophageal cancers (10–30%), but also occurs in ovary, endometrium, head and neck, colon, bladder, and lung cancers, thus representing an important prognostic biomarker but also serving as an almost ideal drug target [7]. When overexpressed, it is pharmaceutically targeted with monoclonal antibodies Trastuzumab and Pertuzumab as well as the combination of the two, with the bioconjugate Ado-Trastuzumab Emtansine, and with diverse tyrosine kinase inhibitors (Neratinib, Afatinib, and Lapatinib).

Aloe-emodin (8-dihydroxy-3-hydroxymethyl-anthraquinone; AE), an anthraquinone isolated from many plant species such as *Cassia occidentalis*, *Rheum palmatum* L. (rhubarb), *Aloe vera*, *Polygonum multiflorum* Thunb, and *Rumex* sp. [8] has attracted great attention for its diverse biological activities [9] and especially for its anticancer activity towards various human cancer cell lines [10,11].

In U937 monoblastic leukemia cells [12], B16-F10 melanoma cells [13], and the MKN45 HCG cell line [14] it has shown an anti-proliferative activity. Besides the antiproliferative and pro-apoptotic effect, AE also behaves as a motility and angiogenesis inhibitor [15,16,17], which demonstrates that the mechanism of action may be different depending on the target cell line [18].

On the other hand, one of the problems encountered in the use of AE stems from its low bioavailability and the presence of a certain degree of toxicity at the level of some tissues and organs, including the liver [9,19]. Furthermore, the phenolic structure can lead to damage at the mitochondrial level through the development of oxidative stress [20].

The advantage of using peptides as transporters of organic molecules resides in many aspects. The peptide-based delivery of bioactive molecules has several advantages over other delivery methods. Peptides also have a good solubility in serum; in addition, their sizes and physico-chemical features make peptides more efficient in penetrating tissues with respect to antibodies. In addition, the recognition specificity is higher for peptides with respect to small molecules, and the current peptide synthesis techniques allow for a lower cost of production compared to humanized antibodies, making them even more popular.

The aim of this study was to design a peptide-based conjugate coupling the molecule of AE with a specific peptide sequence to produce a proof-of-concept molecule with enhanced water solubility, which can specifically target the HER2 receptor that is overexpressed in certain cell populations. To this aim, we have used the human breast adenocarcinoma cell line SKBR3 as a model, checked for its overexpression of the HER2 receptor, and compared its bioconjugate efficacy towards a lung adenocarcinoma cell line (A549).

## 2. Results and Discussion

### 2.1. Design and Synthesis of AE Derivatives and Conjugates

HER2 is considered an ideal target for the specific delivery of anticancer drugs due to its overexpression in several cancer cell lines [6] reported in several studies [21]. For our study, SKBR3 was selected as a model cell line, known to overexpress this kind of receptor [22]. Thus, to obtain a functional peptide capable of specifically recognizing the HER2 receptor, we adopted the sequence LTVSPWY (LTV peptide) obtained by a biopanning phage-display process [23] and described in diverse studies linked to therapeutics and imaging agents for the specific delivery to cancer cells [24,25,26]. Another feature recently shown by this sequence consists of its ability to act as a penetrating sequence, not only specifically targeting SKBR3 human breast cancer cells but also acting as a membrane-penetrating peptide [27].

Following the concept of a multifunctional peptide, we wanted to add other features to the conjugate by adding the PKKKRKV sequence for different functional aims. Firstly, it would improve the nuclear targeting of AE, thus allowing this sequence to correspond to the Simian-virus 40 (SV40) T-antigen NLS, a motif known to specify nuclear location [28]. Indeed, AE (among its different mechanisms) is known to exert an antitumoral effect by inhibiting different steps of the cell cycle [29]; furthermore, it was reported to enter the cell nucleus and interact with DNA [30]. Second, the basic character of this sequence would also facilitate the endosomal escape pathways [31]; in fact, some cationic peptides may alter the endosome’s structure, promoting its disruption through different mechanisms [32] and thus avoiding their transport to the lysosomes and their consequent degradation or extrusion from cells [33,34]. Towards this aim, to support the hypothesis of a potential ability of the PKKKRKV sequence to interact with biological membranes, a predictive trial using the CellPPD web tool (https://webs.iiitd.edu.in/raghava/cellppd/index.html, accessed on 1 June 2022) was carried out. This bioinformatic tool is dedicated to the prediction of cell-penetrating sequences [35], and as a result the PKKKRKV motif was scored as a CPP.

Hence, the two functional sequences were linked by a GPG motif and the resulting sequence PKKKRKVGPGLTVSPWY (CPP-LTV peptide, Table 1) was obtained.

The GPG hinge was inserted to avoid a ulk between the two sequences allowing for a better exploitation of their respective functions. 

The produced compound showed a higher solubility in water (20 mg/mL) or in media used for cell growth, conversely to AE, which is practically water insoluble.

To conjugate AE (**2**) to the peptide moiety, the hydroxymethyl group was modified by adding a glutaryl group to obtain the product (**3**), which was linked to the peptide chain in the last step of peptide synthesis. The ester bond was chosen as a target for the intracellular metabolic transformation of the conjugate by cell esterase enzymes to free the cargo upon cell internalization. The AE derivative (**3**) was achieved as depicted in Figure 1. Aloin (**1**) was used as a convenient starting material to obtain AE.

A similar strategy to link an anthraquinone to a peptide chain by a glutaryl moiety has been reported in [36], where the peptide moiety was represented by the luteinizing hormone-releasing hormone. Recently, another example of modified AE as an ethyl succinate derivative was reported to study its effect on the NLRP3 inflammasome signaling pathway and its possession of anti-inflammatory effects was demonstrated [37]. The choice to use an ester group was based on the high presence of esterases within cells; these enzymes are at the basis of many bioassays to measure cell viability, such as calcein-AM, which requires esterase cleavage of the acetoxymethyl (AM) ester to become fluorescent.

Fluorescein isothiocyanate (FITC) derivatives of the CPP-LTV (**4**) and GPGLTVSPWY (*g*LTV) peptides (**5**) were produced to test the functionality of the designed peptides by flow cytometry and confocal microscopy experiments, as described in the following paragraph.

### 2.2. AE and AE Conjugates Internalization Experiments

To verify the efficiency of the sequences used to promote receptor recognition and internalization, two FITC peptides derivatives were produced, the compound (**4**) (Figure 2) presenting the full-length peptide sequence comprising the two functional motifs (CPP and HER2 recognition), and compound (**5**) (Figure 2), bearing only the receptor recognition sequence *g*LTV. To assess the expression of HER2 in both the cell lines, an initial flow cytometric analysis of the receptor surface labelling was performed (Figure 1). The ratio between the mean fluorescence channel (MFC) of the Ab-labelled cells (Figure 1B,D) and MFC of the control cells (Figure 1A,C) showed that SKBR3 receptor expression was almost three times higher when compared with A549 cells (ratio = 2.80 ± 0.16 versus 1.15 ± 0.18, respectively, from three independent experiments) [23].

Uptake experiments of compounds (**4**) and (**5**) were performed in the SKBR3 and A549 cell lines to assess the extent of their accumulation and intracellular localization by means of flow cytometry and confocal microscopy, respectively. For the flow cytometry measurements, cell suspensions were incubated with the two compounds (5 μM) for 15 and 30 min at 4 °C (Figure 2A, top histogram graphs) and at 37 °C (Figure 2B, top histogram graphs) in order to verify if temperature conditions could influence the uptake process.

The results showed that the two peptides’ derivatives behaved differently with respect to the two cell lines, as the fluorescence signal derived from compound (**4**) was more intense—both at 4 °C and 37 °C—in the SKBR3 and A549 cells than compound (**5**), suggesting its higher uptake inside the cells. In good agreement with the flow cytometry data, the confocal images showed a similar trend (Figure 2A–H).

These results also demonstrated that the presence of a penetrating sequence (PKKKRKV) strongly enhanced the entry of the FITC-peptide, more than doubling the amount of internalized peptide derivative (**5**) in all the conditions. Second, the presence of the LTV sequence, specific for the HER2 receptor, enabled the more efficient targeting of the peptide derivative (**4**) towards the SKBR3 cells. Indeed, these data agree with the flow cytometry results, showing that the lower compound uptake observed in the A549 cells corresponded to the lower HER2 expression.

In addition, the experiments performed at 37 or 4 °C suggested that the uptake process did not depend on the temperature used.

To study the internalization of the peptide conjugate (**6**), confocal microscopy images of SKBR3 and A549 cells were taken at different times upon treatment with 10 μM of the compound (Figure 3). The first set of images was taken in real time cell culture conditions in the first 30 min to visualize the early entry events enabled by the autofluorescence in the green channel (ex 488 nm/em500–535 nm) of the AE moiety of (**6**). As a result, at 15 min, the entry of (**6**) was limited for both cell lines with punctuate spots visible inside the cells (Figure 3), suggesting an endocytosis mechanism of uptake. At 45 min, the green fluorescence was more diffuse in both cell lines with a higher intensity in the SKBR3 cells, indicating a more elevated rate of uptake with respect to A549. Similar to that observed for the FITC-peptides, the higher expression of HER2 in the SKBR3 cells appears to enhance the internalization of the conjugate in this cell line.

A second set of images was taken in the fixed cells after 24 h and 48 h upon conjugate treatment, to visualize the uptake in a more prolonged time lapse (Figure 3C,D,G,H).

Even in this case, differences both in the distribution and amount of fluorescence could be observed between the two cell lines. In the SKBR3 cells, at 24 h, the fluorescence was distributed in the cytoplasmic space, while in the same time lapse in the A549 cells the fluorescence was partially distributed in the cytosol in punctuate structures and partially in the nuclear space. At 48 h, the differences in green channel fluorescence localization were more pronounced. In the A549 cells, fluorescence was mainly distributed in spots with a slightly higher concentration near the plasmatic membrane (Figure 3D). Conversely, in the SKBR3 cells, fluorescence appeared concentrated in the nucleus. In the merged images (Figure 3H), green fluorescence is clearly mixed with H33258 nuclear stain, strongly suggesting a migration of the molecules into the nucleus. The presence of such a dense fluorescence within the SKBR3 cells’ nuclei suggests that AE freed by its cargo upon cytoplasmic esterase’s action may passively enter the nuclei; this event was observed in neuroblastoma cell lines while using two-photon excitation microscopy [30]. The possibility that even the whole conjugate (**6**) may be targeted within the nuclei could not be excluded, as nuclear pores allow for the passive entry of small molecules and peptides [38]. Indeed, the PKKKRKV sequence corresponding to the Simian-virus 40 (SV40) T-antigen NLS is a motif known to specify nuclear location and used in other studies to deliver biomolecules into this cell compartment [39].

### 2.3. Cell Viability Analysis in SKBR3 and A549 Cells

To analyze the effects of the products (**2**), (**3**), and (**6**) on cell viability, MTT tests were performed in SKBR3 and A549 cells treated at 24, 48, and 72 h with different concentrations, (Figure 4A–F). As a result, AE (**2**) induced a strong cytotoxic effect mainly observed with the higher dose at 50 μM in both the cell lines. A reduction in cell viability ranging from 60 to 70% (depending on treatment time) was observed in SKBR3, and from 50 to 60% in the A549 cells, with a major effect at 72 h (Figure 4A,F). The glutaryl-AE derivative (**3**) already showed to be effective already at 5 µM for both the cell lines (Figure 4B,E). The conjugate (**6**) induced a reduction of 40% of cell viability with 5 and 10 μM at 24 h in the SKBR3 cells (Figure 4C); conversely, at the same conditions in the A549 cells, the cell viability reduction was 10–20% (Figure 4F). In both cell lines, the higher concentrations (20 and 50 µM) administered at any treatment time induced a strong reduction in cell viability equal to or higher than 70% (Figure 4C,F). In our experimental models, it is interesting to notice that the glutaryl-derivative (**3**) was more effective in reducing cell viability when compared to AE (**2**). To our knowledge, studies regarding the effects of such a derivative of AE on cancer cells are not reported. These data also show that conjugate (**6**) already showed greater cytotoxic activity than the glutaryl-derivative (**3**) at a concentration of 20 μM; thus, it is more efficient at lower concentrations.

### 2.4. Cytotoxicity Studies of AE and AE Derivatives on NK92 and K562 Cells

AE not only exerts antineoplastic and anti-inflammatory actions, but also inhibits the differentiation and maturation of dendritic cells (DCs), induces the proliferation of T lymphocytes in vitro [40], and affects Natural killer (NK) cell activity in a dose-dependent manner [41]. This suggests that AE has a significant effect on the immune system. NK cells constitute a first line of defense against cancer and can kill target cancer cells. NK cells are the most powerful arm of the innate immune system and play an important role in immunity, alloimmunity, auto-immunity, and the body’s battle against cancer. NK cells are proving to amplify the immune response in molecularly targeted therapies [42]. Since NK cells are considered key components in multidisciplinary cancer therapeutic strategies, we wanted to investigate whether AE and AE derivatives could mediate NK cell activity.

To exclude the possibility that AE, at the conditions to be used in NK92 cell–mediated cytotoxicity experiments, could interfere with cellular viability, the direct effect of the AE on the NK-92 cell line was firstly tested by analyzing apoptosis, proliferation, and growth inhibition. Treatment of NK-92 with the AE and its derivatives (**3** and **6**) did not induce growth inhibition, and neither AE (**2**) induced apoptosis or affected proliferation after 48 h at all the evaluated concentrations, except for the higher dose (50 μM) that produced a cytotoxic effect (Figure 5).

To assay the functional role of AE in mediating NK cells’ killing activity, we performed an in vitro study where NK92 cells were incubated with AE (**2**) and its derivatives (**3** and **6**) using K562 as target cells added to the cultures. The NK92 cell line was treated with increasing AE (**2**), AE-CPP-LTV (**6**), and glutaryl-AE (**3**) concentrations (ranging from 5 to 20 μM) at an effector: target (E:T) ratio of 10:1 and then K562 cells were added to the cultures. As a control, untreated NK92 cells were used. The results showed that the treatment of NK92 cells with AE (**2**) and glutaryl-AE (**3**) induced less NK92-mediated cytotoxicity than the control sample (untreated NK92 cells), while the NK92 cells previously treated with AE-CPP-LTV (**6**) showed similar activity compared with the untreated control, demonstrating that it does not interfere with NK cells’ killing activity and cellular processes (Figure 6).

The bioconjugate (**6**) maintained the NK cells’ activity at control levels; hence, this bioconjugate represents a highly efficient form of AE that avoids the side effect of the natural compound.

From the data collected and in the light of the negative effects of AE on NKs’ function (Figure 6), the ability of the conjugate (**6**) to retain the NK cells’ cytotoxic activity towards cancer cells may simply be due to the presence of a peptide moiety that could reverse the negative effect of AE. Indeed, it is well known that NKs are sensitive to selected peptides that can modulate their cytotoxic action. Peptide binding Class I MHC molecules [43], virus-derived peptides [44], neuropeptides [45], and even nutraceutical peptides [46] are known to exert a positive effect on NKs’ function. The PKKKRKVGPGLTVSPWY peptideis composed of two sequences known to have specific functions related to HER2 recognition (LTVSPWY) and targeting of the cell nucleus (PKKKRKV). To our knowledge, there are no known functions attributable to these or similar sequences related to the modulation of natural killer cells. A further investigation will be needed to verify if this sequence reliably performs the important role of being an activator of NKs.

## 3. Materials and Methods

### 3.1. Aloe-Emodin (AE) Derivatives Synthesis and Characterization

All air- and moisture- sensitive reactions were carried out using dry solvents and under a static atmosphere of nitrogen. All solvents and reagents were of commercial quality and were purchased from Sigma-Aldrich (St. Louis, MO, USA).

Aloin (**1**) from *Curacao aloe* (52% purity) lot SLBC4749V was purchased from Sigma-Aldrich (St. Louis, MO, USA). AE (**2**) was obtained by oxidation of commercial aloin (**1**) with ferric chloride, according to the procedure described by Rychener and Steiger [47]. Synthesis of 5-((4,5-dihydroxy-9,10-dioxo-9,10-dihydroanthracen-2-yl)methoxy)-5-oxopentanoic acid (**3**) was achieved as follows. A sample of Aloe-emodin (500 mg, 1.85 mmol) in dry toluene (20 mL) was treated with glutaric anhydride (300 mg, 2.63 mmol) and with a catalytic amount of *p*-toluenesulfonic acid monohydrate (20 mg, 0.1 mmol). The mixture was heated at reflux under nitrogen atmosphere until complete transformation of the starting Aloe-emodin (**2**) (TLC analysis, 8 h). The reaction was then cooled, and the solvent was removed under reduced pressure. The residue was purified by chromatography using *n*-hexane/AcOEt (8:2–1:1) as eluent to afford the glutaryl derivative **3** (470 mg, 66% yield). Rf of 3 = 0.12 and Rf of aloe-emodin = 0.21 (hexane/ethyl acetate 6:4) [48].

#### Analytical Methods and Characterization of the Products

Nuclear Magnetic Resonance spectroscopy (NMR), ^1^H- and ^13^C-NMR Spectra, and DEPT experiment: CDCl_3_ solutions at RT using a Bruker-AC-400 spectrometer (Billerica, MA, USA) at 400, 100, and 100 MHz, respectively; ^13^C spectra are proton decoupled; and chemical shifts in ppm relative to internal SiMe_4_ (=0 ppm).

TLC: Merck silica gel 60 F254 plates (Merck Millipore, Milan, Italy). Column chromatography: silica gel (Appendix A).

Melting points were measured on a Reichert apparatus (Reichert, Vienna, Austria), equipped with a Reichert microscope, and are uncorrected.

Mass spectra were recorded on a Bruker ESQUIRE 3000 PLUS spectrometer (ESI detector) (Billerica, MA, USA) or by GC-MS analyses. Orange crystal; M.p: 164–166 °C.

^1^H NMR (400 MHz, DMSO-*d*6) δ 11.98 (br s, 3H), 7.84–7.77 (m, 1H), 7.71 (d, *J* = 7.5 Hz, 1H), 7.67 (d, *J* = 1.0 Hz, 1H), 7.38 (d, *J* = 8.3 Hz, 1H), 7.33 (s, 1H), 5.22 (s, 2H), 2.33–2.19 (m, 4H), 1.85–1.75 (m, 2H).

^13^C NMR (100 MHz, DMSO-*d*6) d 191.5 (C), 181.1 (C), 173.9 (C), 172.2 (C), 161.3 (C), 161.3 (C), 146.4 (C), 137.3 (CH), 133.4 (C), 133.2 (C), 124.4 (CH), 122.0 (CH), 119.3 (CH), 117.7 (CH), 115.8 (C), 115.3 (C), 64.1 (CH_2_), 32.5 (CH_2_), 32.5 (CH_2_), 19.8 (CH_2_).

MS (ESI): 407.0 (M+Na^+^); 383.0 (M-1, negative ions) (Appendix A).

### 3.2. Peptides and Conjugates Synthesis and Characterization

Resins, N-α-Fmoc-L-amino acids, and building blocks used during chain assembly were purchased from Iris Biotech GmbH (Marktredwitz, Germany). Ethyl cyanoglyoxylate-2-oxime (Oxyma) was purchased from Novabiochem (Darmstadt, Germany), N,N′-dimethylformamide (DMF) and trifluoroacetic acid (TFA) were from Carlo Erba (Rodano, Italy). N,N′-diisopropylcarbodiimide (DIC), dichloromethane (DCM) and all other organic reagents and solvents, unless stated otherwise, were purchased in high purity from Sigma-Aldrich (Steinheim, Germany). All solvents for solid-phase peptide synthesis (SPPS) were used without further purification. HPLC grade acetonitrile (ACN) and ultrapure 18.2 Ω water (Millipore-MilliQ) were used for the preparation of all buffers for liquid chromatography.

Peptides were assembled by stepwise microwave-assisted Fmoc-SPPS on a Biotage ALSTRA Initiator+ peptide synthesizer, operating at a 0.1 mmol scale. Activation of entering Fmoc-protected amino acids (0.3 M solution in DMF) was performed using 0.5 M Oxyma in DMF/0.5 M DIC in DMF (1:1 molar ratio), with a 5-equivalent excess over the initial resin loading. Coupling steps were performed for 7 min at 75 °C. Fmoc- deprotection steps were performed by treatment with a 20% piperidine solution in DMF at room temperature (1 × 10 min). Following each coupling or deprotection step, peptidyl-resin was washed with DMF (4 × 3.5 mL). Upon complete chain assembly, resin was washed with DCM (5 × 3.5 mL) and gently dried under a nitrogen flow. For the synthesis of FITC-labelled derivatives, resin-bound peptides were incubated overnight with a solution of FITC-reagent (2 eq.), DIEA (4 eq.) in DMF at room temperature and then washed with DMF.

For the synthesis of FITC-labelled derivatives, resin-bound peptides were incubated overnight with a solution of FITC-reagent (2 eq.), DIEA (4 eq.) in DMF at room temperature and then washed with DMF.

For the coupling of Aloe-emodin derivative (**3**) to peptide CPP-LTV to obtain the final product (**6**), compound (**3**) was activated using 0.5 M Oxyma in DMF/0.5 M DIC in DMF (1:1:1 molar ratio), with a 1.5 equivalent excess over the initial resin loading. Coupling was performed overnight at room temperature. Post-synthesis work-up was identical for all the synthesized peptides/derivatives. Resin-bound peptides were treated with an ice-cold TFA, TIS, water, and thioanisole mixture (90:5:2.5:2.5 *v/v/v/v*, 4 mL) for 2 h; then, the cleavage mixture was added dropwise to ice-cold diethyl ether (40 mL) to precipitate the crude peptide, collected by centrifugation. Peptides were then purified by RP-HPLC using as eluents the following A: 97.5% H_2_O, 2.5% ACN, 0.7% TFA, HPLC eluent B: 30% H_2_O, 70% ACN, and 0.7% TFA.

#### Analytical Methods and Characterization of the Products

Analytical RP-HPLC was performed on a Shimadzu Prominence HPLC (Shimadzu) using a Shimadzu Shimpack GWS C_18_ column (5-micron; 4.6 mm i.d. × 150 mm). Analytes were eluted using a binary gradient of mobile phase A (100% water and 0.1% trifluoroacetic acid) and mobile phase B (30% water, 70% acetonitrile, and 0.1% trifluoroacetic) using the following chromatographic method: 10% B to 100% B in 14 min; flow rate—1 mL/min. Preparative RP-HPLC was performed on a Tri Rotar-VI HPLC system (JASCO) using a Phenomenex Jupiter C18 column (10 micron; 21.2 mm i.d. × 250 mm) using the following chromatographic method: 0% B to 90% B in 45 min; flow rate—14 mL/min. Pure RP-HPLC fractions (>95%) were combined and lyophilized. Electro-spray ionization mass spectrometry (ESI-MS) was performed using a Bruker Esquire 3000+ instrument equipped with an electro-spray ionization source and a quadrupole ion trap detector (QITD).

### 3.3. Cell Lines

Human breast adenocarcinoma cell line SKBR3 and human non-small cell lung cancer cell line A549 were grown in RPMI 1640 (Euroclone, Milan, Italy). Both cell lines were obtained from American Type Culture Collection (ATCC, Manassas, VA, USA) growing as monolayer in media supplemented with 10% heat-inactivated FBS (HyClone™ Fetal Bovine Serum (U.S.), Characterized, Marlborough, MA, USA), 1% l-glutamine, and 1% penicillin (100 U/mL)/streptomycin (100 U/mL) (all chemicals were purchased from Euroclone, Italy) and subcultured prior to confluence using trypsin/EDTA.

NK-92, an interleukin-2 dependent Natural Killer cell line, was a kind gift from Prof. L. Moretta (Tumor Immunology Research Unit, IRCCS Ospedale Pediatrico Bambino Gesù, Rome, Italy) who obtained the cells from ATCC repository (ATCC; CRL-2407). Human chronic myeloid leukemia K562 cells were purchased from ATCC. Cells were cultured in a basic medium (BM) that was constituted by RPMI-1640 supplemented with 10% FBS, penicillin/streptomycin, and l-glutamine as reported above. NK-92 in vitro growth is dependent on the presence of recombinant IL-2 (100 IU/mL, CellGS, St. Louis, MO, USA) [49]. All cells in this study were maintained at 37 °C in a humidified 5% CO_2_ atmosphere.

### 3.4. Flow Cytometry

#### 3.4.1. Analysis of HER2 Expression on SKBR3 and A549 Cell Lines

To determine cell surface HER2 expression, flow cytometric analysis was carried out on cell suspensions (10^6^ cells/mL) obtained by incubating adherent cultures with EDTA and trypsin solutions. Cells were incubated for 30 min at 4 °C with Erb2 (HER2) monoclonal antibody (Thermofisher, Waltham, MA, USA) working dilution 1:100). For negative controls, cells were incubated with isotype immunoglobulins (mouse IgG1) at the same dilution. After washing with ice-cold PBS containing 10 mM NaN_3_ and 1% BSA (Sigma), cells were incubated for 30 min at 4 °C with a secondary goat–anti-mouse IgG Alexa-Fluor 488-conjugate (Invitrogen, Eugene, OR, USA) at a working dilution of 1:50. After two washings in cold PBS, cells were labelled with propidium iodide (PI, Sigma, and final concentration 40 µg/mL) and immediately analyzed on a LSRII flow cytometer (Becton and Dickinson, Franklin Lakes, NJ, USA), equipped with a 5 mW, 488 nm, air-cooled argon ion laser and a Kimmon HeCd 325 nm laser. Fluorescence emissions were collected through a 530 and 570 nm band-pass filters for Alexa-Fluor and PI signals, respectively. At least 10,000 cells per sample were acquired in log mode. Only PI-negative cells were considered for analysis. The fluorescence intensity values were expressed as mean fluorescence channel (MFC) using the FACS Diva software (Becton Dickinson). To quantify the HER2 expression level, the ratio between the MFC of labelled cells and the MFC of control ones was calculated from three independent experiments.

#### 3.4.2. Cellular Uptake of AE Derivatives in SKBR3 and A549 Cell Lines

Cell suspensions (10^6^ cells/mL) were incubated with compounds (**4**) and (**5**) (Figure 2), which were previously prepared in phosphate buffer saline (PBS) to obtain a stock solution of 1 mM, at the final concentration of 5 μM in growth medium for 15 and 30 min. At the end of incubations (performed at 4 and 37 °C to assess the role of temperature in the uptake process), samples were washed in cold PBS, labelled with PI, and immediately acquired on LSR II flow cytometer. At least 10,000 cells per sample were acquired in log mode. Only PI-negative cells were considered for analysis. The ratios between the mean fluorescence channel (MFC) of compound-treated cells and that of control ones (cells treated with vehicle alone) were calculated to quantify the increase of FITC fluorescence emission (related to compound uptake) with respect to the autofluorescence signal of control samples (Diva software). Statistical analysis was performed by the Student’s *t*-test (2-tailed) to assess differences between means of data analyzed. The *p*-values are indicated in the figure legends. The statistical analysis was performed with GraphPad Prism 8 (Graph Pad Software, San Diego, CA, USA).

#### 3.4.3. Analysis of Apoptosis and Cell Proliferation in NK-92 Cells

Apoptosis was quantified using FITC-conjugated annexin V (AV) according to the manufacturer’s protocol (Marine Biological Laboratory, Woods Hole, MA, USA). Annexin V-FITC staining in NK-92 cell lines was determined after 48 h in the absence or presence of different amounts of Aloe-emodin (9 mM stock solution in DMSO) ranging from 6.25 to 50 μM. Proliferation was evaluated by measuring Ki-67 nuclear antigen expression using FITC-labeled anti-human Ki-67 mAb according to the manufacturer’s protocol (BD Biosciences, San Jose, CA, USA). Acquisition was performed on a FACSCalibur flow cytometer (BD Biosciences) and at least 10,000 events per sample were run. Data were analyzed using the Cell Quest Pro software (BD Biosciences) and expressed as mean ± standard deviation from 2 independent experiments.

### 3.5. Cell Viability Assay

The compounds (**2**), (**3**), and (**6**), were evaluated for cell viability effects using the MTT assay on SKBR3 and A549 cells at 24, 48, and 72 h at different concentrations. Cell lines were seeded into 96-well microtiter plates (Corning, Inc., Corning, NY, USA) at a density of 1 × 10^4^ cells/well. After 24 h, cells were exposed to increased concentrations of compounds (previously prepared in PBS to obtain a stock solution of 1 mM) in cell culture medium. After an incubation time of 24, 48, and 72 h, media were removed and MTT (Sigma, Deisenhofen, Germany) diluted in culture media was added to each well at 0.5 mg/mL. Plates were incubated for 2 h at 37 °C in 5% CO_2_ incubator; unreacted dye was removed, and cells were solubilized in DMSO (Merck, Darmstadt, Germany). Absorbance was read at 570 nm by an ELISA plate reader (Biorad). Data were expressed as cell metabolic activity, which was calculated as the absorbance value of cells treated with different compounds/absorbance value of control (cells incubated with vehicle alone) cells * 100%. All experiments were performed in triplicate, and all data were expressed as the mean ± standard deviation. One-way ANOVA with Dunnett’s multiple comparisons test was performed with respect to the controls (**** *p* < 0.0001). The statistical analysis of differences was performed with GraphPad Prism 8.

### 3.6. Cytotoxicity Assay

To determine NK-92’s cell cytotoxicity, CytoTox 96 non-radioactive cytotoxicity assay (Promega, Madison, WI) was used, based on the colorimetric detection of the released enzyme LDH. NK-92 cells were harvested, washed, counted, and diluted at the concentration of 1 × 10^6^ cells/mL in BM with 10% FCS and 100 μ/mL of IL-2. Effector cells were then seeded in 24-well Costar plates (Costar, Rochester, NY, USA) and maintained for 48 h at 37 °C alone or in the presence of 20 µM, 10 µM, and 5 µM of Aloe-emodin (**2**), glutaryl derivative of AE (**3**), and AE-CPP-LTV (**6**). Preliminary dose response and time course experiments showed that AE (**2**) should be used at a dose maximum of 20 µM at 48 h of culture to obtain the highest detectable changes in the absence of toxic effects. Then, the NK-92 cells treated with and without Aloe (cells incubated with vehicle alone) were collected, extensively washed with warm RPMI-1640, and seeded at a density of 1 × 10^6^ cells/mL, serially diluted, ranging routinely from 1 × 10^5^ to 6.25 × 10^3^ cells, in round-bottom 96-well Costar plates (Costar, Rochester, NY, USA). Target cells were harvested, washed, resuspended to 1 × 10^5^ cells/mL in BM, added, and incubated with effector cells for 4 h at 37 °C employing incresingincreasing effector/target cell ratio ranging from 0.625:1 to 10:1 in triplicate. A total of 50 μL of supernatants were assayed for LDH activity according to the manufacturer’s instruction. Controls for spontaneous and maximum LDH release in effector and target cells were also assayed. The calculation of cytotoxicity percentage was as follows: (Experimental—Effector Spontaneous—Target spontaneous)/(Target Maximum—Target Spontaneous) × 100. Data are expressed as mean ± standard deviation from 4 independent experiments. Statistical analysis was performed by the Student’s *t*-test (2-tailed) to assess differences between means of data analysed. The *p*-values are indicated in the figure legends. The statistical analysis was performed with GraphPad Prism 8.

### 3.7. Growth Inhibition Assay

NK-92 cell lines in exponential phase of growth were collected, extensively washed with warm RPMI-1640, and seeded (in triplicate) in 96-well microtiter Costar plates (Costar, Rochester, NY, USA) at a density of 5 × 10^3^ cells/mL. The cells were cultured in BM containing recombinant IL-2(100 IU/mL). In the growth inhibition assays, NK92 cells were maintained for 24, 48, and 72 h at 37 °C alone or in the presence of 20, 10, 5, and 1 µM of Aloe-emodin, glutaryl derivatives of AE and AE-CPP-LTV (6). For all above-described experiments, cell survival was determined by WST-8 assay (Orangu™ Cell Counting SolutionWST-8 cell proliferation kit (Aurogene, Rome, Italy), after 24, 48, and 72 h treatment at 37 °C in 5% CO_2_. Cells were treated with different compounds/absorbance values of control (cells incubated with vehicle alone) cells * 100%. Statistical analysis was performed by the Student’s *t*-test (2-tailed) to assess differences between means of data analyzed. The *p*-values are indicated in the figure legends. The statistical analysis of differences was performed with GraphPad Prism 8.

### 3.8. Confocal Microscopy

For confocal laser scanning microscopy, SKBR3 and A549 cells cultured on 12 mm diameter coverslips were treated with 5 and 10 μM of Aloe-emodin AE (**2**) and AE-CPP-LTV peptide (**6**) for 24 and 48 h. For “in vivo” images cells (2 × 10^4^) were seeded in µ-Slide 8 Well high ibiTreat (Ibidi, Grafelfing, Germany) for 24 h. Peptides were used at 10 μM and nuclei were stained with Hoechst 33342 (Sigma, St Louis, MO, USA). Then, coverslips were fixed with 4% paraformaldehyde for 30 min at room temperature and permeabilized for 5 min with 0.5% Triton X-100. For actin detection, cells were stained with TRITC-phalloidin (Sigma, Milan, Italy) at 37 °C for 20 min. For nuclei detection, cells were stained with Hoechst 33258 (Sigma, St. Louis, MO, USA) at 37 °C for 15 min. At the end, coverslips were mounted with Vectashield (Vector Laboratories) and images were carried out by a Zeiss LSM980 apparatus, using a 63×/1.40 NA oil objective, and excitation spectral laser lines at 405, 488, and 543. Image acquisition and processing were carried out using the Zeiss Confocal Software Zen 3.3 (Blue edition). Several cells for each labeling condition were analyzed and representative results were shown.

## 4. Conclusions

In this study, the possibility of joining the potential of AE as an anticancer drug and its targeted delivery towards specific cell lines was explored. This aim was pursued by producing a conjugate molecule composed of Aloe-emodin linked to a properly designed multifunctional peptide moiety (Figure 7).

The confocal images of the two cell lines upon conjugate treatment show the presence in the first few minutes and even later (24 h and 48 h) of well-defined punctuate structures, thus suggesting the hypothesis that the conjugate may be internalized through an endocytosis process. The recognition of the LTVSPWY sequence, which is highly specific for HER2 (**1**), may induce the formation of endosomes (**2**) containing the conjugate bound to the receptor. This does not exclude the possibility that the conjugate may enter for passive translocation due to the presence of the LTVSPWY sequence, recently recognized as a penetrative sequence itself [27], and to the PKKKRKV also predicted to be a CPP sequence (**3**) corresponding to these features of the peptide moiety, and this may also determine the endosomal escape that will release the conjugate into the cytosol where esterases will hydrolyze the ester bond of the conjugate (**4**) itself, thus freeing the AE moiety in the cytosol and then for passive diffusion into the nucleus (**5**). Alternatively, the conjugate may be led towards the nucleus due to the PKKKRKV motif.

The data obtained using the SKBR3 and A549 cell lines as models showed that (i) the cell viability reduction induced by the conjugate was higher with respect to the natural compound in both cell lines; (ii) the uptake of peptide-conjugated AE was higher in the SKBR3 cells compared to A549, due to the different expression of HER2 in the two cell lines; (iii) conversely to AE, the conjugate exerted a preservative effect on the NK cells’ function. These results demonstrate that the strategy used was overall successful, thereby paving the way for a more efficient and specific system of the delivery of AE to be translated in a precision medicine framework.

## Data Availability

All the data produced in this study are reported in this article. The primary data files are available from the corresponding authors upon reasonable request.

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
