# Peer review of "Peptide-Mediated Targeted Delivery of Aloe-Emodin as Anticancer Drug"

_molecules, 2022, doi:10.3390/molecules27144615_

Round 1
Reviewer 1 Report
This paper prepared several aloe-emodin derivations modified with functional peptides and evaluated the cell specificity, cell viability effects and immune cell response. The following points should be addressed.
1. Line 116, “Indeed, AE among different mechanisms, is known to exert its antitumoral effect by inhibiting different steps of cell cycle [29]”. This effect of Compound 2, 3 and 6 to inhibit cell cycle should be examined.
2. In Figure 3, what is “in vivo”condition?
3. Why did the conjugate (6) keep the NK cells cytotoxic activity towards cancer cells while Compound 2 and 3 not? More detailed discussion shuld be provided.
4. The 4.4.1~4.4.3 should be 3.4.1~3.4.3.
Reviewer 2 Report
In this work Stringaro and coworkers present the development of peptide-based conjugate enhancing delivery and anticancer potential of aloe-emodin. First of all, I would like to congratulate the Authors since they performed tremendous work and the results obtained are also quite optimistic, especially taking into account enhanced cytotoxicity potential and the selectivity of synthesized conjugate.
The manuscript contains the results of complementary experiments, all steps are explained sufficiently, it is easy to follow, and is very interesting for the readers, but it needs to be revised by the Authors:
- Scheme 1 – in figure capture are presented numbers of compounds, whereas there is lack of numbers identifying the chemicals directly near the presented structures;
- Line 173 – it seems that in the sentence there is lack of word “higher”;
- Part 2.2, line 185 - please explain the rationale for usage of compounds 4 and 5 at 5 µM concentration; did the Authors checked the cytotoxicity of these compounds? These data will allow to explain that the lower ratio of compound 5 uptake is not connected with its cytotoxicity (on photos C, D, G, H is observed lower number of cells)
-Figure 2, 3– please, enlarge the photos and add a scale bar;
-lines 189, 209, 244, etc – the term “in vivo” is related to animal based experiment; here the cell lines were used therefore I suggest to use term “real time cell culture conditions” or something similar; please unify this within the whole manuscript (including figure captures)
- Line 209 – why 10 µM of compound 6 was used? The cytotoxicity studies present in Figure 4 C, F its cytotoxicity even at 5 µM concentration.
- Lines 214-216 – I do understand the sentence, but to be more precise it needs to be modified;
- Line 274 – usually in this type of experiments the data is calculated using the equation:
metabolic activity = abs of cells treated with compound/abs of control cells * 100%
Please explain the rationale for presented type of data processing (“Data were expressed as absorbance values relative to untreated control cells, considered as 100%.”) . Please explain, what were the solvents used for studied compounds; what was the stock solution concentration? What was used as the cell “control” in quantitative data?
- Figure 5 capture – please add the numbers of studied compounds, i.e. 2, 3, 6.
- Frankly speaking, all figure captures are presented in different styles and require modifications in this regard. The usage of chemical numbers, i.e. 2,3,6 instead of A, B, C and additional explanations will make the text more easy to follow.
In summary I recommend the minor revision of the manuscript and I would like to read the answers from Authors and revised manuscript.
Round 2
Reviewer 1 Report
Statistical analysis should be performed in Figure 2, Figure 4, Figure 5 and Figure 6.
